# Epigenetic inheritance of circadian period in clonal cells

Yan Li[1], Yongli Shan[1], Gokhul Krishna Kilaru[1], Stefano Berto[1], Guang-Zhong Wang[1†], Kimberly H Cox[1], Seung-Hee Yoo[1‡], Shuzhang Yang[1], Genevieve Konopka[1], Joseph S Takahashi[1,2]*

[1]Department of Neuroscience, Peter O'Donnell Jr. Brain Institute, University of Texas Southwestern Medical Center, Dallas, United States; [2]Howard Hughes Medical Institute, University of Texas Southwestern Medical Center, Dallas, United States

*For correspondence:
joseph.takahashi@utsouthwestern.edu

Present address: [†]CAS Key Laboratory of Computational Biology, CAS-MPG Partner Institute for Computational Biology, Shanghai Institute of Nutrition and Health, Shanghai Institutes for Biological Sciences, University of Chinese Academy of Sciences, Chinese Academy of Sciences, Shanghai, China; [‡] Department of Biochemistry and Molecular Biology, The University of Texas Health Science Center at Houston, Houston, United States

Competing interests: The authors declare that no competing interests exist.

**Abstract** Circadian oscillations are generated via transcriptional-translational negative feedback loops. However, individual cells from fibroblast cell lines have heterogeneous rhythms, oscillating independently and with different period lengths. Here we showed that heterogeneity in circadian period is heritable and used a multi-omics approach to investigate underlying mechanisms. By examining large-scale phenotype-associated gene expression profiles in hundreds of mouse clonal cell lines, we identified and validated multiple novel candidate genes involved in circadian period determination in the absence of significant genomic variants. We also discovered differentially co-expressed gene networks that were functionally associated with period length. We further demonstrated that global differential DNA methylation bidirectionally regulated these same gene networks. Interestingly, we found that depletion of DNMT1 and DNMT3A had opposite effects on circadian period, suggesting non-redundant roles in circadian gene regulation. Together, our findings identify novel gene candidates involved in periodicity, and reveal DNA methylation as an important regulator of circadian periodicity.

## Introduction

Circadian oscillations maintain daily rhythms to control multiple physiological and behavioral processes, including metabolism, cell growth, immune response, and the sleep-wake cycle. Disruptions of the circadian clock have been linked with various disease processes and aging (*Takahashi et al., 2008*; *Kondratova and Kondratov, 2012*). Circadian oscillations display remarkable fidelity in their periodicity even in the absence of environmental cues. This precision of the internal biological clock arises from a complex gene network. In mammals, the core of this network is composed of an autoregulatory transcriptional negative feedback loop involving *Clock*, *Bmal1*, *Per1/Per2*, and *Cry1/Cry2*, and there are additional feedback loops interlocked with the core (*Takahashi et al., 2008*; *Mohawk et al., 2012*; *Takahashi, 2017*). Interestingly, although the cell-autonomous clock is ubiquitous, individual cells often do not maintain a perfect 24 hr circadian period, and within cell populations there are heterogeneous autonomous oscillations with a broad distribution of period length (*Nagoshi et al., 2004*; *Welsh et al., 2004*; *Leise et al., 2012*). The heterogeneity in intrinsic period of hypothalamic suprachiasmatic nucleus (SCN) neurons confers important functions of phase liability and phase plasticity (*Welsh et al., 1995*; *Liu et al., 1997*; *Ko et al., 2010*; *Mohawk et al., 2012*). However, it is still unclear how heterogeneous circadian periodicity is established and maintained under physiological conditions, or how much of this heterogeneity is heritable.

The origin of heterogeneity is complex, but may be driven by genetic variation, epigenetic modifications, and/or transcriptional noise (*Jaenisch and Bird, 2003*; *Raser and O'Shea, 2005*; *Raj and van Oudenaarden, 2008*; *Burrell et al., 2013*; *Kelsey et al., 2017*; *Cavalli and Heard, 2019*;

*Liu et al., 2019*). We have recently shown that nonheritable noise is the predominant source of inter-cellular variation in circadian period within clonal cell lines (*Li et al., 2020*). However, it is still unclear what heritable factors contribute to period variation among different clonal cells. DNA methylation has been recognized as a chief contributor to gene expression states, and it is essential for mammalian embryonic development, with genome-wide methylation patterns changing during differentiation (*Greenberg and Bourc'his, 2019*). There are three canonical cytosine-5 DNA methyltransferases that catalyze the addition of methylation marks. DNMT3A and 3B, the de novo methyltransferases, set up DNA methylation patterns during early development. Once established, DNMT1 will copy those patterns onto the daughter strand during DNA replication ensuring methylation maintenance (*Jaenisch and Bird, 2003*). DNMT dysfunction has been associated with various diseases, and DNMT-deficient mice exhibit embryonic lethality (*Greenberg and Bourc'his, 2019*). Numerous studies have supported the role of DNA methylation in gene silencing; however, more recent work suggests that DNA methylation can also be involved in transcriptional activation (*Rinaldi et al., 2016*; *Yin et al., 2017b*; *Harris et al., 2018*; *Lyko, 2018*). Interestingly, despite high fidelity in mitotic inheritance, DNA methylation is variable across individuals, tissues, and cell types (*Jaenisch and Bird, 2003*; *Jones, 2012*; *Varley et al., 2013*). Thus, we hypothesized that differential DNA methylation could contribute as a heritable factor underlying heterogeneous circadian oscillations in clonal cell lines.

Here, by examining phenotype-associated high-throughput multi-omics profiles in clonal cell populations, we identified and validated a pool of novel candidate genes regulating circadian period length and uncovered complex gene co-expression networks highly enriched in stress response and metabolic pathways. We next explored the origins of heterogeneous gene expression and found differences in global DNA methylation patterns that were associated with both silencing and activation of differentially expressed genes. Using gene knockdown studies, we also found that DNMT1 and DNMT3A have opposite effects on period length. Together, our findings demonstrate the important role of DNA methylation in the regulation of circadian period.

## Results

### Heritable circadian periodicity in clonal cell lines

To assess cellular phenotypic heterogeneity, we utilized an immortalized mouse ear fibroblast cell line carrying a PER2::LUC*sv* bioluminescence reporter generated from *Per2::lucSV* knockin mice (*Chen et al., 2012*; *Yoo et al., 2017*). We recently showed that these cells express persistent, robust, and cell-autonomous circadian oscillations over a 2 week period. Moreover, clonal cell lines generated from the parent culture had period distributions similar to those seen with single cells, indicating that circadian period is a heritable phenotype (*Figure 1A–B*; *Li et al., 2020*). Here, we used the clonal cell lines to address the underlying molecular mechanism for heterogeneous circadian periodicity. To examine the stability of this heritability, twenty clonal cell lines were randomly selected and cultured continuously for 20 passages and tested for circadian period every five passages. Although two-way ANOVA revealed significant effects ($p<0.01$) of both cell line and passage, there was no interaction ($p=0.09$). Moreover, cell line was the dominant source of variation (74.70%), while passage only contributed 2.64% of the total variation. Multiple comparisons within each clonal cell line across passages identified a significant difference (adjusted $p<0.05$) for only ~5% of comparisons (11 out of 200), which is consistent with 5% false positive rate. These results indicate that circadian period of clonal cell lines is stable and transmissible for at least 20 cell passages (*Figure 1C*).

### Transcriptomics identifies novel gene candidates determining period length

To explore potential underlying mechanisms, we selected two groups of clonal cell lines from the two tails of the period distribution (*Table 1*, 5 short period (SP) and five long period (LP) clones) (*Li et al., 2020*) and performed RNA-seq analysis (*Figure 2—source data 1*). We compared their transcriptomic profiles and identified 5,137 period-correlated differentially expressed (DE) genes, with 2,782 genes upregulated and 2,355 genes downregulated in the LP group (*Figure 2A*, *Figure 2—source data 1*). To narrow down the target pool further and identify candidate genes more directly responsible for periodicity differences, we selected four additional groups of subclones

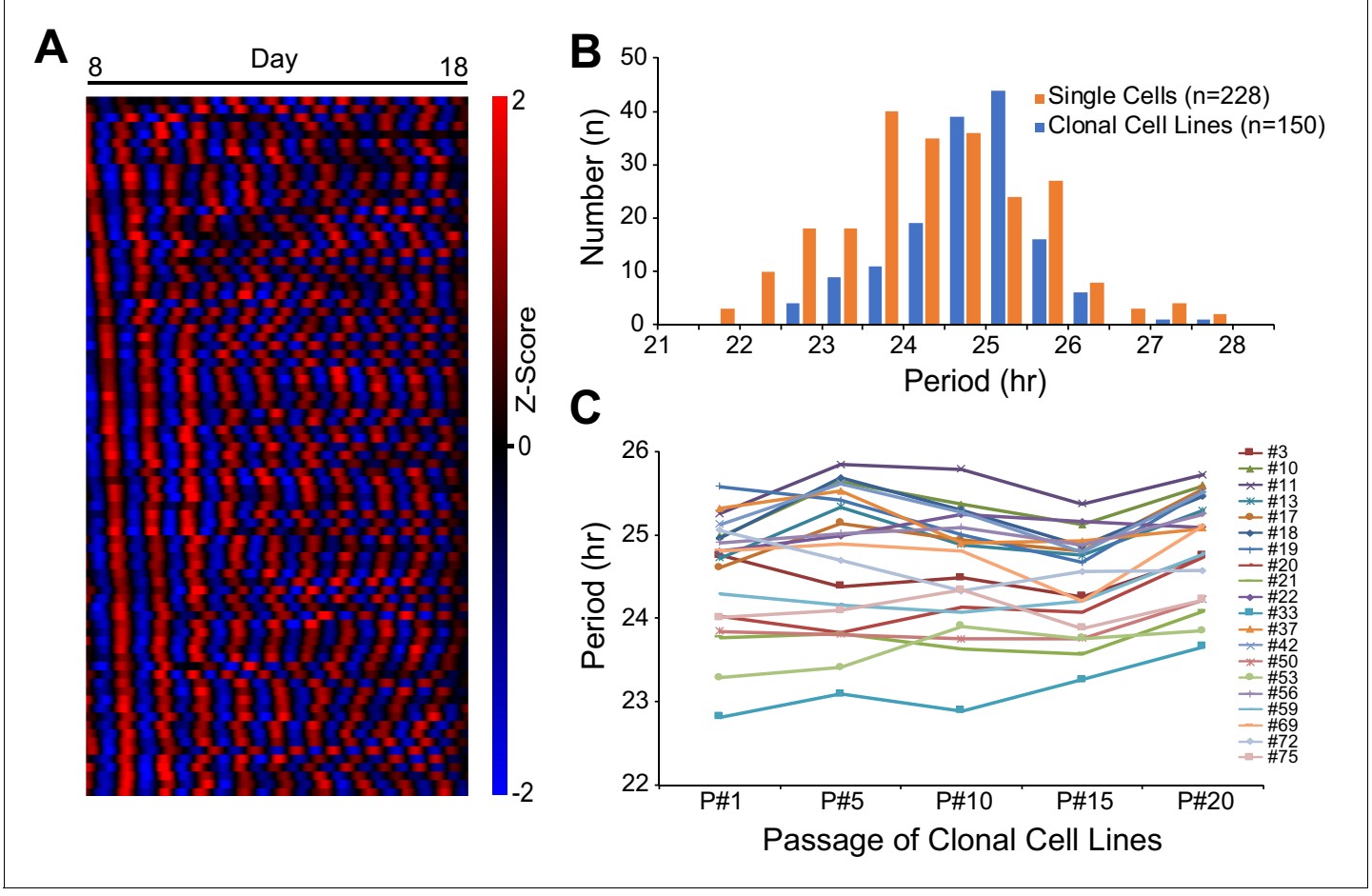

**Figure 1.** Heritable Circadian Periodicity in Clonal Cell Lines. (**A**) Heatmap showing circadian oscillations of 83 single cells from parent culture tracked continuously for 10 days (sorted by phase at day 8). (**B**) Histogram showing circadian period distributions of single cells compared to clonal cell lines generated from the same parent culture. Single cells: 24.38 ± 1.20 hr (mean ± SD), ranged 21.55–27.82 hr. Clonal cell lines: 24.81 ± 0.83 hr, ranged 22.76–27.65 hr. Clonal cell lines were measured as a whole culture. Data are replotted from *Li et al., 2020* and presented as averages from ≥3 experiments. (**C**) Periods of individual clonal cell lines of different generations. Periods were analyzed for the whole culture at passages 1, 5, 10, 15, and 20. Data are presented as averages from ≥3 experiments.

established from two representative clonal cell lines with different periods: a shorter period subgroup and a longer period subgroup from short period clone#33 (SSP and LSP), or long period clone#114 (SLP and LLP), respectively (*Figure 2B*; *Li et al., 2020*). These subclones and the original 10 clonal cell lines constituted a continuous period spectrum beneficial for identifying period-correlated genes (*Figure 2C*, *Table 1*).

We identified 535 additional period-correlated DE genes from subclones originating from SP clone#33 and 1,352 additional DE genes from subclones originating from LP clone#114 (*Figure 2D–E*, *Figure 2—source data 1*). By comparing the three RNA-seq datasets, 67 overlapping DE genes were identified (*Figure 2F*). From these, we selected 14 genes based on the strength of the correlation between their expression and circadian period length from all 88 samples and performed knockdown experiments to validate their function in circadian periodicity. Out of 7 positively correlated DE genes, knockdown of *Ak3* and *Trim3* significantly shortened period, whereas knockdown of *Cpeb1*, *Lrrfip1*, *Rbfa*, and *Dars* lengthened period (*Figure 3A–C*, *Figure 3—source data 1*). Out of 7 negatively correlated DE genes, knockdown of *Ipo13* and *Tmem165* significantly lengthened period, whereas *Slc8a3*, *Jun*, *Med23*, and *Cpa4* knockdown shortened period (*Figure 3D–F*, *Figure 3—source data 1*). Knockdown of two other genes, *Eif4e2* and *Rfx5*, did not alter period length. We also examined the effect of knockdown of five representative genes in 10 clonal cell lines and found that they all showed the same period alterations as that seen in the parent culture

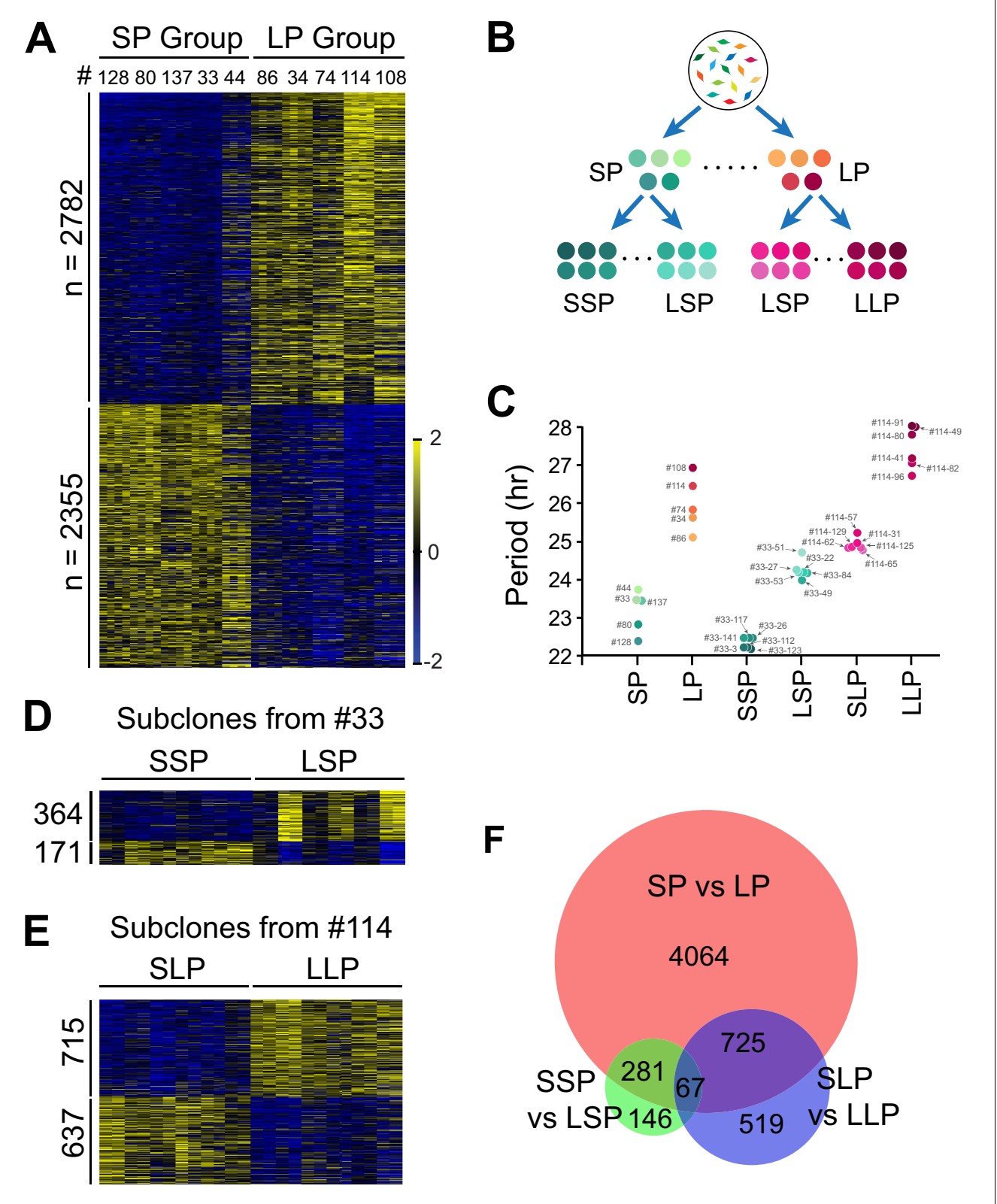

**Figure 2.** Differentially Expressed Genes Correlated with Circadian Period Heterogeneity. (A) Heatmap of 5,137 period-correlated differentially expressed (DE) genes identified between two groups of clonal cell lines. SP: short period. LP: long period. Clones were sorted based on period length. Clone IDs shown at top. Four columns for each clone indicate two time-points, two replicates. From left to right: replicate1_T1, replicate1_T2, replicate2_T1 and replicate2_T2. Color scale represents z-score. (B) Experimental scheme for establishing subgroups of clonal cell lines carrying

*Figure 2 continued on next page*

*Figure 2 continued*

different circadian periods. SSP: shorter period subgroup from short period clone. LSP: longer period subgroup from short period clone. SLP: shorter period subgroup from long period clone. LLP: longer period subgroup from long period clone. (C) Scatter plot showing period length of different groups. Each dot represents a clonal cell line. (D) Heatmap of 535 period-correlated DE genes identified between two groups of subclones derived from SP clone#33. Each group include six subclones sorted based on period length. From left to right: #33–123, #33–112, #33–3, #33–26, #33–117, #33–141, #33–49, #33–84, #33–22, #33–53, #33–27, #33–51. Two columns for each sample indicate two time-points. (E) Heatmap of 1,352 period-correlated DE genes identified between two groups of subclones derived from LP clone#114. Each group include six subclones sorted based on period length. From left to right: #114–65, #114–125, #114–62, #114–129, #114–31, #114–57, #114–96, #114–82, #114–41, #114–80, #114–49, #114–91. Two columns for each sample indicate two time-points. (F) Area-proportional Venn diagram comparing DE genes identified above. For more information, see *Figure 2—source data 1*.

The online version of this article includes the following source data for figure 2:

**Source data 1.** RNA-seq of All Clonal Cell Lines; List of Period-correlated DE Genes.

demonstrating the overall consistency of the gene knockdowns on circadian period (*Figure 3G*, *Figure 3—source data 1*). These results suggest that multiple genes function together to determine circadian period length and that there were no unique (clone-specific) effects on the direction (long or short) of the period changes. Since the majority of the DE genes identified here have never been reported as having effects on circadian period, these data provide a new pool of candidate genes functioning in circadian periodicity.

## Large-scale gene networks are associated with period heterogeneity

Because functionally related genes are usually co-expressed (*Heyer et al., 1999*), we further characterized the period-correlated DE genes by examining their co-expression patterns. Using weighted

**Table 1.** Period of Clonal Cell Lines

| Group | Clone # | Period (hr)[*] | STDEV | Group | Clone # | Period (hr)[*] | STDEV |
|---|---|---|---|---|---|---|---|
| SP | #128 | 22.35 | 0.13 | LP | #86 | 25.07 | 0.12 |
| | #80 | 22.78 | 0.10 | | #34 | 25.60 | 0.50 |
| | #137 | 23.42 | 0.21 | | #74 | 25.80 | 0.50 |
| | #33 | 23.43 | 0.07 | | #114 | 26.43 | 0.15 |
| | #44 | 23.70 | 0.10 | | #108 | 26.90 | 0.14 |
| SSP | #33–123 | 22.15 | 0.43 | SLP | #114–65 | 24.74 | 0.20 |
| | #33–112 | 22.18 | 0.18 | | #114–125 | 24.80 | 0.14 |
| | #33–3 | 22.20 | 0.21 | | #114–62 | 24.81 | 0.21 |
| | #33–26 | 22.43 | 0.21 | | #114–129 | 24.82 | 0.16 |
| | #33–117 | 22.43 | 0.27 | | #114–31 | 24.93 | 0.35 |
| | #33–141 | 22.44 | 0.38 | | #114–57 | 25.19 | 0.13 |
| LSP | #33–49 | 23.95 | 0.27 | LLP | #114–96 | 26.69 | 0.68 |
| | #33–84 | 24.13 | 0.27 | | #114–82 | 27.02 | 0.47 |
| | #33–22 | 24.16 | 0.13 | | #114–41 | 27.16 | 0.45 |
| | #33–53 | 24.16 | 0.20 | | #114–80 | 27.77 | 0.44 |
| | #33–27 | 24.22 | 0.32 | | #114–49 | 27.97 | 0.99 |
| | #33–51 | 24.68 | 0.13 | | #114–91 | 28.00 | 0.66 |

* Average of ≥3 experiments.

SP: short period group.

LP: long period group.

SSP: shorter period subgroup from short period clone#33.

LSP: longer period subgroup from short period clone#33.

SLP: shorter period subgroup from long period clone#114.

LLP: longer period subgroup from long period clone#114.

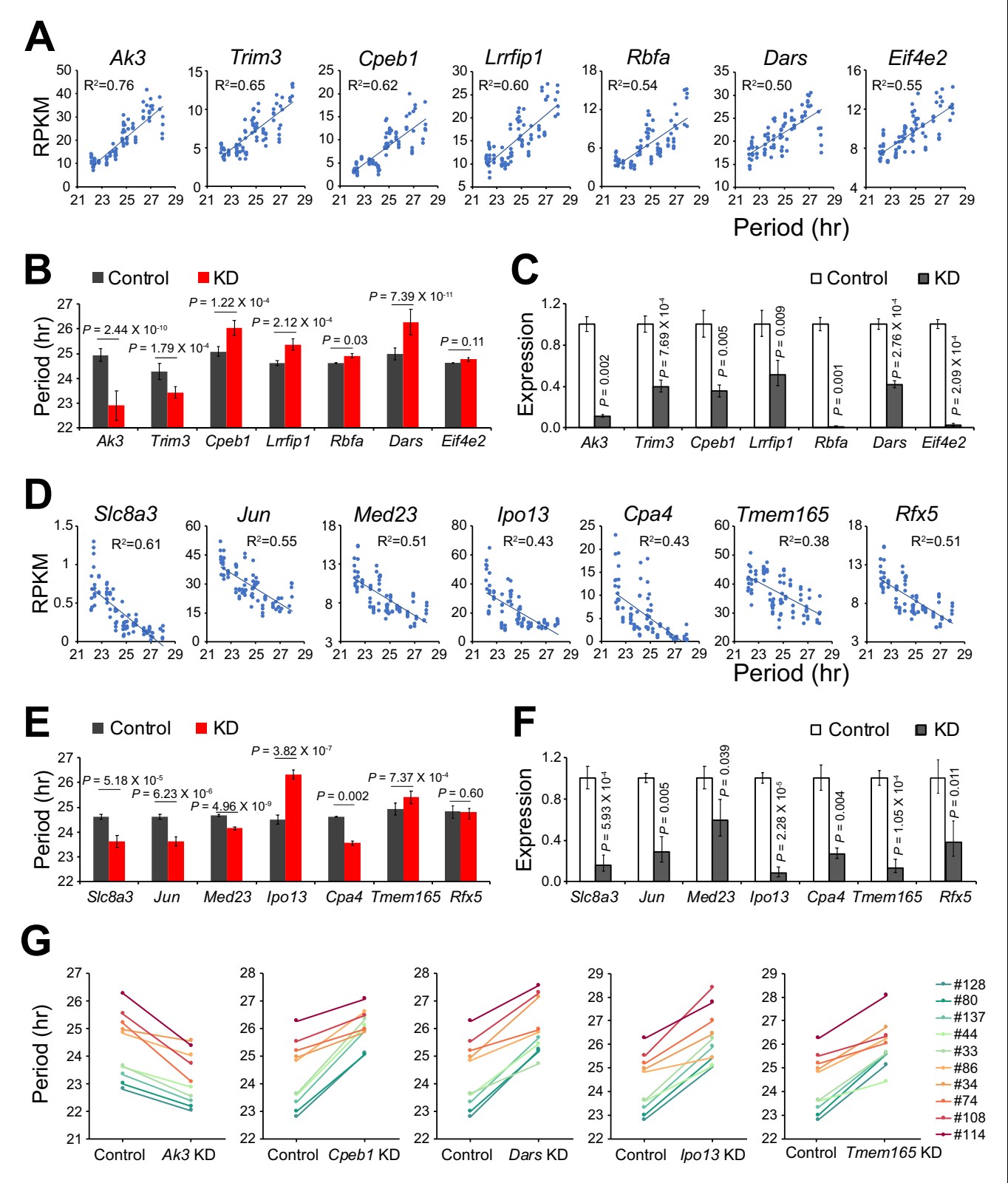

**Figure 3.** Validation of Novel Candidate Genes Regulating Circadian Periodicity. (A) Scatter plot of seven novel candidate genes showing positive correlation between gene expression and period length across all 88 samples. (B) Histogram comparing periods after knocking down seven positively correlated genes in parent culture. n ≥ 3 for each gene. Error bar indicates SD. (C) QPCR results showing knockdown efficiency of seven positively correlated genes. Error bar indicates SD. (D) Scatter plot of seven novel candidate genes showing negative correlation between gene expression and

*Figure 3 continued on next page*

Figure 3 continued

period length across all 88 samples. (E) Histogram comparing periods after knocking down seven negatively correlated genes in parent culture. n ≥ 3 for each gene. Error bar indicates SD. (F) QPCR results showing knockdown efficiency of seven negatively correlated genes. Error bar indicates SD. (G) Knockdown of five representative candidate genes in 10 clonal cell lines. For more information, see *Figure 3—source data 1*.

The online version of this article includes the following source data for figure 3:

**Source data 1.** Gene Knockdown in Parent Culture; Gene Knockdown in 10 Clonal Cell lines.

gene co-expression network analysis (WGCNA), we generated 31 modules from the 10 clonal cell lines RNA-seq data (*Figure 4A*, *Figure 4—source data 1*). Several modules exhibited significant enrichment for period-correlated DE genes. Blue, lightgreen, green and darkred modules were enriched for positively correlated DE genes, while salmon, pink, red, and darkgreen modules were enriched for negatively correlated DE genes (*Figure 4B*).

Ingenuity pathway analysis (IPA) revealed stress response signaling pathways and metabolic pathways were associated with the period-correlated DE genes, suggesting their important roles in circadian periodicity (*Figure 4C*, *Figure 4—source data 1*). IPA analysis of the correlated modules also revealed overlapping functional pathways. For example, the blue module is highly enriched for DE genes, and is also enriched for the EIF2 signaling pathway, which has been recently shown to regulate circadian period, consistent with the predicted elevated translational activity in LP group (*Pathak et al., 2019*; *Figure 4C*). To validate these results further, we used two different small molecules to activate the EIF2 signaling pathway in parent culture and observed significantly shortened period, consistent with what has been previously reported (*Pathak et al., 2019*; *Figure 4D*). In addition, the darkred module was enriched for the mTOR signaling pathway; the green and salmon modules were enriched for the protein ubiquitination pathway; and the pink module was enriched for NRF2-mediated oxidative stress response pathway (*Figure 4—source data 1*). Interestingly, all three of these pathways have been shown to be functional in circadian periodicity, further confirming the functional importance of the co-expressed gene networks (*Stojkovic et al., 2014*; *Ramanathan et al., 2018*; *Wible et al., 2018*). Further analysis of Protein-protein Interactions (PPI) revealed that co-expressed DE genes were also physically interconnected. For example, within the blue module there were several different tightly linked clusters, including those enriched for ribosomal RNA processing, protein ubiquitination, nucleotide and amino acid metabolism, and mRNA splicing, emphasizing the blue module as a transcriptional/translational related gene network (*Figure 4E*). Taken together, our results suggest that period heterogeneity is regulated by changes in large-scale functional gene co-expression networks.

## Global DNA methylation contributes to gene Co-expression networks

To explore whether there was a genetic basis for heterogeneous gene expression, we performed whole-exome sequencing on SP clone#33 and LP clone#114. Interestingly, only four annotated genes carrying unique variants were identified (*Supplementary file 1*), but 2 of them are not expressed (*Figure 2—source data 1*), and none of them have known circadian functions, suggesting that somatic mutations are unlikely to underlie the heterogeneous period distributions.

Cell-to-cell variability is also partially heritable via epigenetic modifications such as DNA methylation (*Jaenisch and Bird, 2003*; *Jones, 2012*). To assess the contribution of DNA methylation in heterogeneous circadian periodicity, we used reduced representation bisulfite sequencing (RRBS) to explore DNA methylation profiles and their correlation with the period-correlated transcriptomes. Using 1,000 bp tiling windows genome-wide, we identified 16,520 significant differentially methylated regions (DMRs). Importantly, none of the core clock genes, even the few that were differentially expressed in the parental lines, had coding mutations or differential DNA methylation, except for a small DMR spanning ~10 nucleotides located in exon 1 of *Per1* (*Table 2*). Of the DMRs found, 62% (10,212 DMRs) were up-regulated, whereas 38% (6,308 DMRs) were down-regulated in the SP group (*Figure 5A*, *Figure 5—source data 1*). 6055 genes were annotated as DMR-associated with DMRs falling in either the gene body or 5 kb upstream of the transcription start site (TSS), and of these, 1,315 DMR-associated genes overlapped with period-correlated DE genes (*Figure 5B*). Interestingly, for period-correlated DE genes associated with DMRs, in addition to negative correlations, we also observed positively correlated DMRs, indicating both repression and enhancement of functional

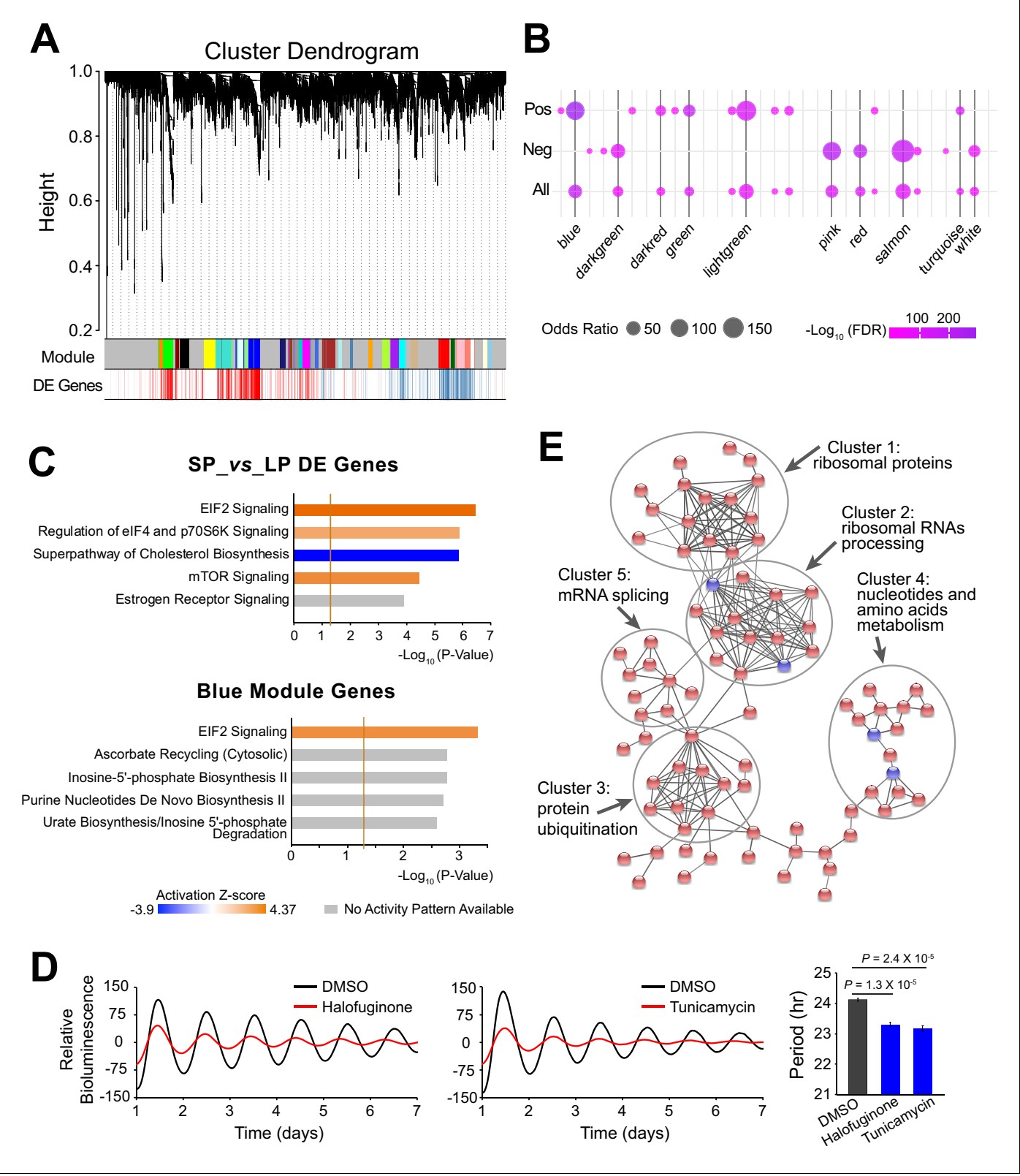

**Figure 4.** Large-scale Co-expressed Gene Networks Associated with Period-correlated DE Genes. (A) Gene co-expression modules identified via WGCNA for 10 clonal cell lines. Each branch in the dendrogram on top represents a cluster of highly correlated genes. Thirty-one modules were identified and marked by colors on the horizontal bar. Bottom row shows period-correlated DE genes identified in *Figure 2a*. Red bars indicate positive correlation (Pearson correlation coefficient >0.5). Blue bars indicate negative correlation (Pearson correlation coefficient <−0.5). (B) Bubble plot

*Figure 4 continued on next page*

*Figure 4 continued*

showing enrichment of period-correlated DE genes in each module. Rows from top to bottom indicate enrichment of positively correlated, negatively correlated, and all period-correlated DE genes, respectively. Dark grey bars highlight the top 10 enriched modules. (C) Top five Ingenuity pathways associated with 5,137 period-correlated DE genes comparing SP and LP groups (top) or blue module (bottom). Brown threshold line refers to p-value=0.05. (D) Two small molecule activators of EIF2 signaling pathway, halofuginone and tunicamycin, significantly shortened circadian period in parent culture. Error bar indicates SD. (E) The main PPI network of genes in blue module. Disconnected nodes were hidden. Red indicates period-correlated DE genes. Line thickness indicates confidence. For more information, see *Figure 4—source data 1*.

The online version of this article includes the following source data for figure 4:

**Source data 1.** WGCNA Module Lists and Enrichment Analysis Results; IPA Canonical Pathway Analysis of DE Genes and WGCNA Modules.

gene expression by DNA methylation (*Figure 5C–D*) as reported by others (*Jones, 2012*; *Rinaldi et al., 2016*; *Yin et al., 2017b*; *Harris et al., 2018*).

The overall clustering pattern of the methylomes resembled that of the transcriptomes, indicating an important role for global DNA methylation in regulating the co-expressed genes (*Figure 5—figure supplement 1*). We examined the modules enriched for period-correlated DE genes and found that several hub genes were regulated by differential DNA methylation. For example, the hub gene of the blue module, *Htatip2*, which exhibited the same expression pattern of the module eigengene (*Figure 6A–B*), was hypermethylated at the promoter region and repressed in the SP group (*Figure 6C–D*). On the contrary, *Parvb* and *Rftn1*, two hub genes from negatively correlated modules, were hypermethylated and repressed in the LP group (*Figure 6A–D*). Except for these negative correlations, some genes with hypermethylation in the gene body or enhancer showed enhanced expression levels (*Figure 6—figure supplement 1*), supporting recent findings that DNA methylation in these regions may activate gene expression (*Jones, 2012*; *Rinaldi et al., 2016*; *Yin et al., 2017b*). To validate the function of DMR-associated DE genes further, we also performed gene knockdown experiments in two different clonal cell lines. Knockdown of *Htatip2* and *Dusp18* in LP clone#114 significantly shortened period, whereas knockdown of *Rftn1* in SP clone#128 significantly lengthened circadian period (*Figure 6E*), consistent with predictions that deficiency of *Htatip2* shortens circadian period possibly by activating the AKT/mTOR signaling pathway (*Yin et al., 2017a*;

**Table 2.** Summary of Sequencing Results of Clock Genes

| Gene | DE gene (SP vs LP) | Fold change (LP/SP) | Adjusted P-Value | DE gene (SSP vs LSP) | DE gene (SLP vs LLP) | Coding mutation | DMR (SP vs LP) |
|---|---|---|---|---|---|---|---|
| *Clock* | Yes | 0.83 | 6.30E-07 | No | No | No | No |
| *Bmal1* | No | N/A | N/A | No | No | No | No |
| *Per1* | Yes | 1.49 | 4.14E-05 | No | No | No | Yes |
| *Per2* | Yes | 1.74 | 0.05 | No | No | No | No |
| *Per3* | No | N/A | N/A | No | No | No | No |
| *Cry1* | No | N/A | N/A | No | No | No | No |
| *Cry2* | No | N/A | N/A | No | No | No | No |
| *Dbp* | No | N/A | N/A | No | No | No | No |
| *Npas2* | No | N/A | N/A | No | No | No | No |
| *Fbxl3* | No | N/A | N/A | No | No | No | No |
| *Fbxl21* | No | N/A | N/A | No | No | No | No |
| *Nr1d1* | No | N/A | N/A | No | No | No | No |
| *Nr1d2* | Yes | 0.63 | 1.56E-04 | No | No | No | No |
| *Csnk1a1* | Yes | 1.29 | 5.38E-03 | No | No | No | No |
| *Csnk1d* | No | N/A | N/A | No | No | No | No |
| *Csnk1e* | Yes | 1.16 | 0.05 | No | No | No | No |
| *Csnk2a1* | No | N/A | N/A | No | No | No | No |
| *Csnk2a2* | No | N/A | N/A | No | No | No | No |

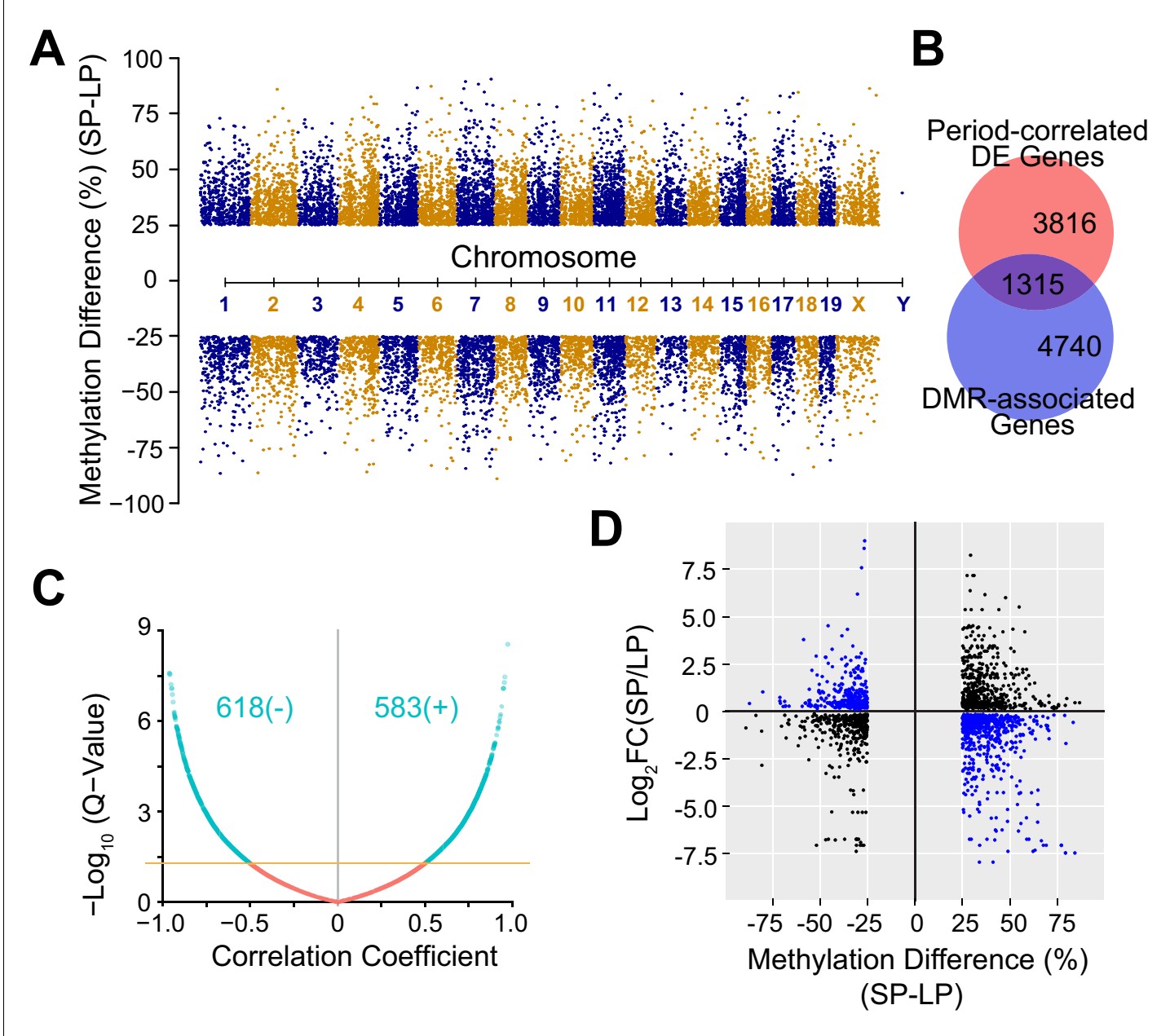

**Figure 5.** DNA Methylation Landscape Associated with Heterogeneous Circadian Periods. (**A**) Manhattan plot showing 16,520 significant DMRs with 1 kb bins. 10,212 DMRs were up-regulated and 6,308 DMRs were down-regulated in the SP group. (**B**) Area-proportional Venn diagram showing overlap between period-correlated DE genes and DMR-associated genes comparing SP and LP groups. Red indicates 5,131 period-correlated DE genes. Blue indicates 6,055 genes associated with significant DMRs. Overlapping area represents 1,315 period-correlated DE genes associated with significant DMRs. (**C**) Volcano plot showing correlation coefficients (Pearson's r) between gene expression and DMR methylation for 1,315 DMR-associated DE genes. X axis indicates Pearson's r. Yellow line indicates q-value = 0.05. Turquoise indicates all significant correlations, including 618 negative and 583 positive correlations. (**D**) Quadrant plot showing relationship between gene expression and DNA methylation. There were 1,915 significant DMRs associated with 1,315 period-correlated DE genes. X axis indicates methylation difference of the associated DMRs. Y axis indicates fold change of averaged gene expression. Blue indicates negative correlations. Black indicates positive correlations. See also *Figure 5—figure supplement 1* and *Figure 5—source data 1*.

The online version of this article includes the following source data and figure supplement(s) for figure 5:

**Source data 1.** DMRs Comparing SP and LP Groups.
**Figure supplement 1.** Clustering of Methylomes Resembles Transcriptomes.

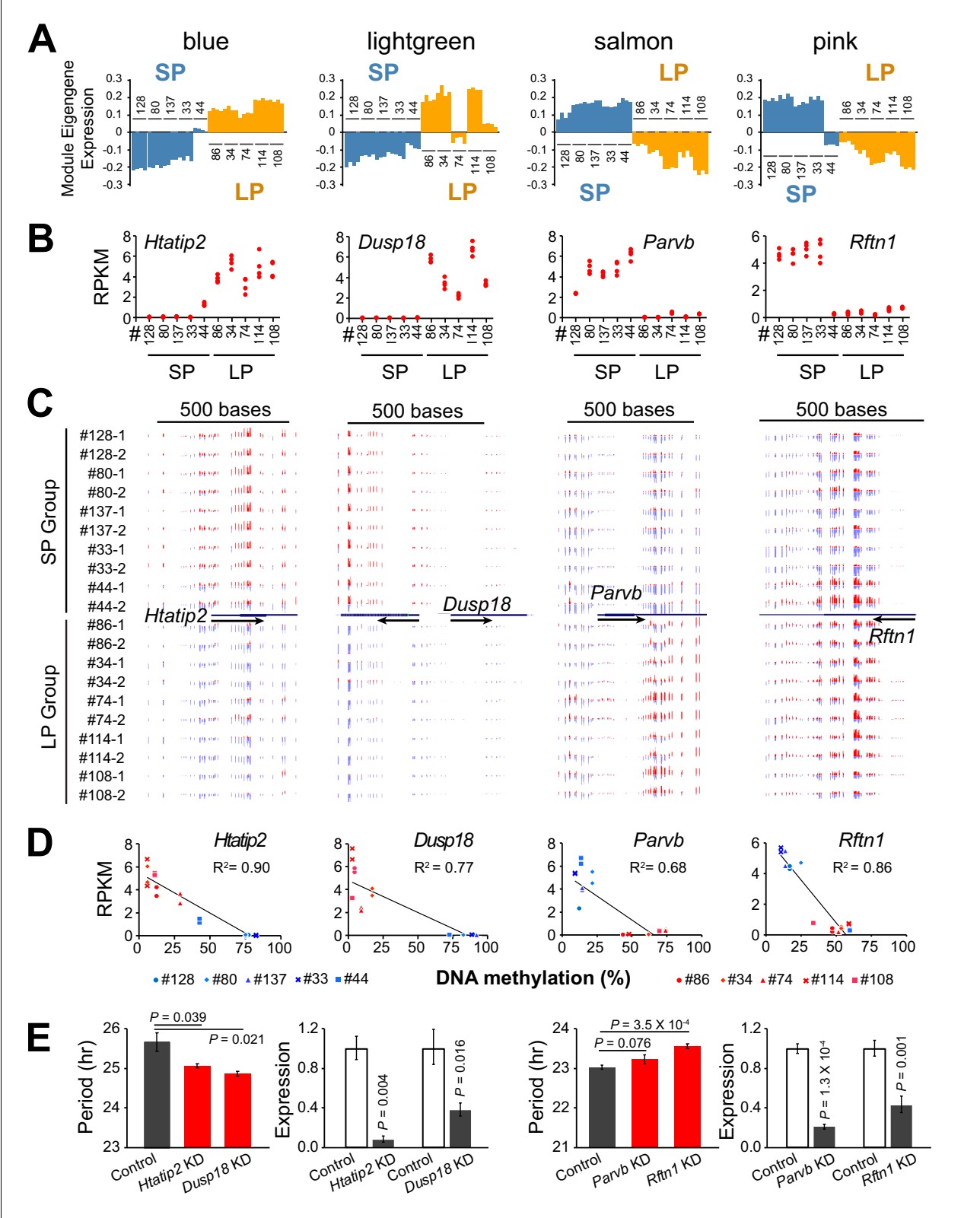

**Figure 6.** DNA Methylation Regulates Expression of Hub Genes from Period-associated Modules. (**A**) Eigengene expression patterns in four top period-associated modules. Blue and lightgreen modules are enriched in positively correlated DE genes with up-regulated gene expression in LP group. Salmon and pink modules are enriched in negatively correlated DE genes with down-regulated gene expression in the LP group. (**B**) Gene expression plot for hub genes of four modules shown in **a**): *Htatip2*, *Dusp18*, *Parvb* and *Rftn1*. Each clonal cell line includes four data points (two time-

*Figure 6 continued*

points, two replicates). (C) UCSC genome browser view of DNA methylation at promoter regions of the four hub genes. Red indicates methylated C. Blue indicates unmethylated C. Arrows indicate the direction of transcription. Track height from left to right: 50,–50; 100,–80; 45,–60; 70,–80. (D) Scatter plot showing correlation between gene expression and DNA methylation at the promoter regions of the four hub genes. X axis indicates average methylation level of the associated DMRs for each clonal cell line. Y axis indicates gene expression level of two replicates at timepoint T1. Trendline and $R^2$ of Pearson correlation coefficient are shown. Blue indicates SP group. Red indicates LP group. DMR loci for each gene are listed as below: *Htatip2*: chr7:49759001–49760000; *Dusp18*: chr11:3894001–3896000. *Parvb*: chr15:84232001–84233000; *Rftn1*: chr17:50190001–50191000. (E) Knockdown effect of four DMR associated DE genes in different clonal cells. Left: *Htatip2* and *Dusp18* knockdown in LP clone#114. Right: *Parvb* and *Rftn1* knockdown in SP clone#128. n ≥ 3 for each experiment. Error bar indicates SD. See also *Figure 6—figure supplement 1*.

The online version of this article includes the following figure supplement(s) for figure 6:

**Figure supplement 1.** Example Genes Associated with Positively Correlated DMRs.

*Ramanathan et al., 2018*), and that hypomethylation and upregulated expression of *Dusp18* lengthens circadian period, possibly by inhibiting the SAPK/JNK signaling pathway (*Wu et al., 2006*; *Chansard et al., 2007*; *Yoshitane et al., 2012*).

## Opposite effects of different DNMTs on circadian period

To assess the role of DNA methylation in circadian periodicity further, we manipulated global DNA methylation either by knocking down DNA methyltransferases or by applying small molecule inhibitors. Interestingly, deficiency of *Dnmt1* significantly shortened period length, whereas knockdown of *Dnmt3a* slightly, but significantly, lengthened period (*Figure 7A–B*). *Dnmt1* and *Dnmt3a* knockdown in the ten clonal cell lines showed the same overall results, suggesting that DNA methylation affects circadian periodicity in the same way in all clones tested (*Figure 7C*). As pharmacological validation, administration of SGI-1027, which selectively induces degradation of the DNMT1 protein (*Datta et al., 2009*), significantly shortened period, while administration of zebularine, which induces significant reduction of both DNMT1 and DNMT3A (*Billam et al., 2010*; *You and Park, 2012*), lengthened period (*Figure 7D*). Drug administration in primary MEF cells with PER2::LUC*sv* and NIH3T3 cells carrying an E2-box-luc reporter also revealed similar results (*Figure 7E*). Taken together, these findings suggest that different DNA methyltransferases contribute to the regulation of circadian periodicity, likely via different mechanisms.

## Discussion

Using clonal cell analysis, we show that the heterogeneity of single-cell circadian periodicity is heritable and stable for at least 20 cell passages. The heritability of circadian period is consistent with an epigenetic mechanism, likely mediated by DNA methylation. By analyzing gene expression profiles of multiple clonal cell lines with different circadian periods, we identified groups of differentially expressed genes that were significantly correlated with period length. Although a few core clock genes were differentially expressed in parental cultures, there were no significant differences in these genes among subclones, suggesting they are not responsible for the period heterogeneity seen in these homogeneous cell populations. By comparing subclones, we narrowed down the common candidate gene list and further validated that 86% of the novel candidates regulated circadian period using gene knockdown assays. While some of these genes had effects on period length that were aligned with our predictions, others had effects counter to our expectations which were probably masked in the complex gene networks. Overall, our results are consistent with the hypothesis that period is determined by the ensemble interactions of many genes that can either shorten or lengthen period individually. Importantly, the vast majority of the DE genes identified here have never been reported as having effects on circadian period. Thus, we have provided a new pool of candidate genes functioning in circadian periodicity.

We also provide evidence that the genome-wide DNA methylation landscape underlies much of the complex gene networks. Multiple hub genes of period-correlated modules were under the regulation of DNA methylation, showing remarkable coherence in DNA methylation, gene expression, and circadian phenotype. The similar clustering patterns of transcriptomes and methylomes further suggested an important role of DNA methylation in shaping circadian period heterogeneity through regulating large-scale gene networks. Previous studies have linked DNA methylation of core clock

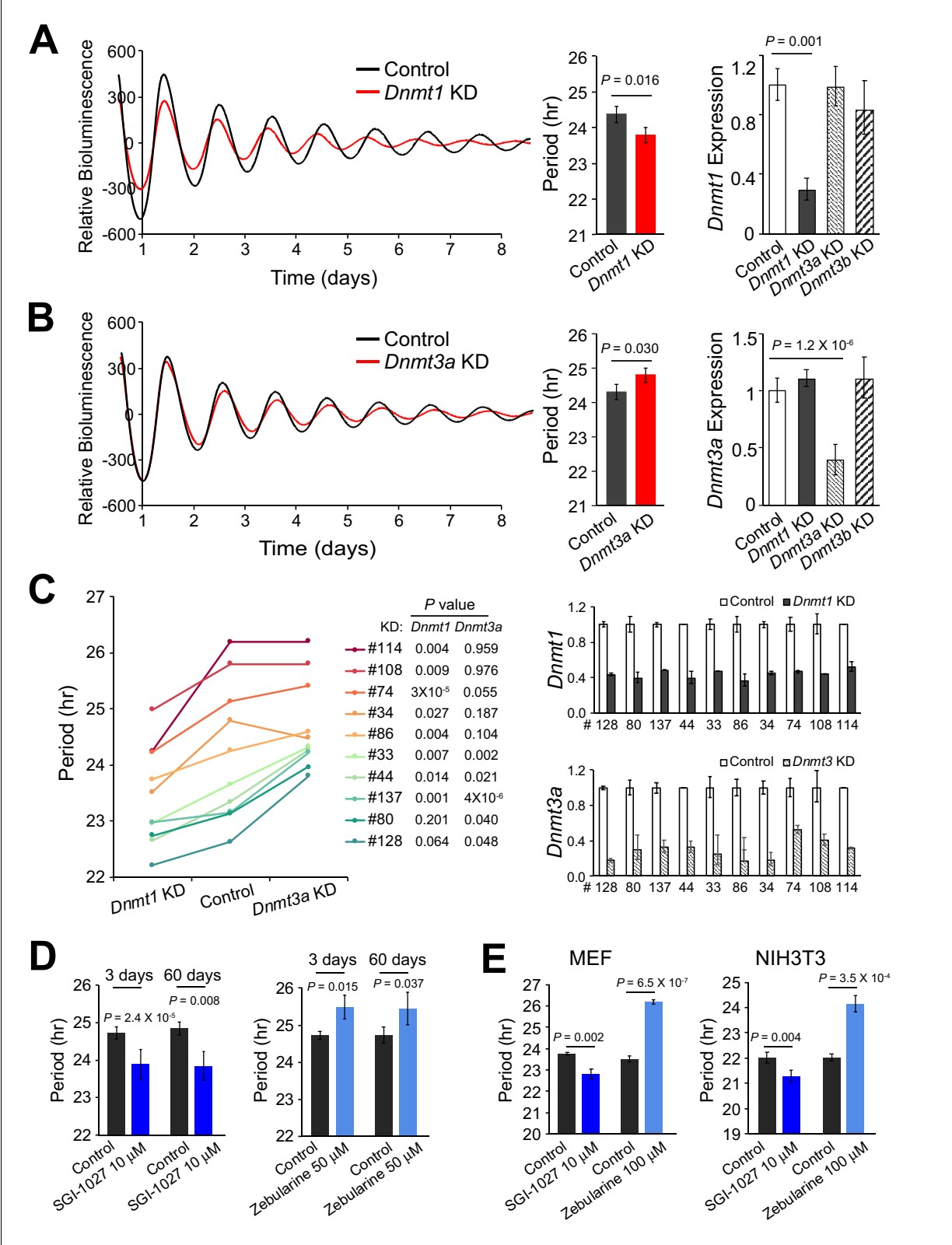

**Figure 7.** Deficiency of Different DNMTs Reveal Opposite Effects on Circadian Period. (**A**) Effects of knockdown of DNMT1 in parent culture. Left: baseline-subtracted LumiCycle traces of a typical experiment; middle: comparison of average period from ≥3 experiments; right: qPCR results showing knockdown efficiency. Error bar indicates SD. (**B**) Effects of knockdown of DNMT3A. (**C**) Effects of knockdown of DNMT1 and DNMT3A in 10 clonal cell lines. Left: comparison of period length from ≥3 experiments. Right: qPCR results showing knockdown efficiency. Error bar indicates SD. (**D**)

*Figure 7 continued on next page*

*Figure 7 continued*

Administration of different DNA methyltransferase inhibitors in parent culture altered period length. Left: SGI-1027. Right: zebularine. n $\geq$ 3 for each experiment. Error bar indicates SD. (E) Administration of different DNA methyltransferase inhibitors in primary mouse embryonic fibroblasts (MEFs) or NIH3T3 cells altered period length. Left: MEF. Right: NIH3T3. n $\geq$ 3 for each experiment. Error bar indicates SD.

genes with different diseases (*Joska et al., 2014*; *Peng et al., 2019*); however, the results presented here have revealed how global DNA methylation can regulate circadian clock function via genome-wide changes in gene expression. Our whole exome sequencing failed to detect significant coding mutations, further supporting the role of differential DNA methylation in establishing circadian heterogeneity. However, we cannot rule out that genetic variation in regulatory regions, or other epigenetic modifications could be involved. Additional experiments will help to understand better the full array of underlying mechanisms regulating circadian period.

We observed both negatively and positively correlated DMRs in almost equal proportions, indicating both repression and activation of gene expression by DNA methylation and supporting the revised view of the functions of DNA methylation (*Greenberg and Bourc'his, 2019*). In addition, we found that knockdown of DNMT1 and DNMT3A had opposite effects on circadian period. It is not surprising that DNMT1 knockdown alters period length, since it is the methyltransferase responsible for DNA methylation maintenance through mitotic inheritance (*Jones, 2012*). However, as DNMT3A is responsible for de novo DNA methylation, it is less clear how its knockdown affects circadian period. One possibility is that DNMT3A is also involved in transcriptional activation associated with active enhancers (*Rinaldi et al., 2016*; *Lyko, 2018*). Another possibility is that some genes might undergo dynamic demethylation and de novo methylation since both *Tet2* and *Tet3* are expressed at comparable levels to *Dnmt3a* in our cellular system (*Oh et al., 2018*; *Oh et al., 2019*; *Figure 2— source data 1*). Additional studies targeting DNMT1 and DNMT3A may help to explain the functions of different DNMTs in circadian regulation.

In conclusion, our findings have identified a novel pool of candidate genes involved in circadian period regulation, and have revealed the important role of DNA methylation underlying circadian period heterogeneity by bidirectionally regulating large-scale gene co-expression networks. Our study not only expands the knowledge about circadian clock regulation, but also may benefit epigenetic research by providing multiple candidate genes repressed or activated by DNA methylation.

# Materials and methods

## Key resources table

| Reagent type (species) or resource | Designation | Source or reference | Identifiers | Additional information |
|---|---|---|---|---|
| Cell line (*M. musculus*, male) | *Per2::lucSV* EF | (*Chen et al., 2012*; *Yoo et al., 2017*) | | Immortalized mouse ear fibroblast cells carrying PER2::LUCsv bioluminescence reporter |
| Cell line (*Human*) | HEK293T | ATCC | CRL-3216 | |
| Cell line (*M. musculus*) | *Per2::lucSV* MEF | This paper | | Primary mouse embryonic fibroblast cells carrying PER2::LUCsv bioluminescence reporter |
| Cell line (*M. musculus*) | NIH3T3/E2LB | This paper | | NIH3T3 cells carrying *Per2* E-box (E2)-driven luciferase bioluminescence reporter |
| Recombinant DNA reagent | pLKO.1-TRC | (*Moffat et al., 2006*) Addgene | Plasmid #10878 | |
| Commercial assay or kit | RRBS kit | Diagenode | Cat#C02030033 | |
| Other | RNA-seq for all clonal cell lines | This paper | | https://www.ncbi.nlm.nih.gov/geo/query/acc.cgi?acc=GSE132663 |

*Continued on next page*

*Continued*

| Reagent type (species) or resource | Designation | Source or reference | Identifiers | Additional information |
|---|---|---|---|---|
| Other | RRBS-seq for 10 clonal cell lines | This paper | | https://www.ncbi.nlm.nih.gov/geo/query/acc.cgi?acc=GSE132665 |
| Other | Exome sequencing for clone#33 and #114 | This paper | | https://www.ncbi.nlm.nih.gov/sra/PRJNA548837 |
| Software, algorithm | FastQC | other | | https://www.bioinformatics.babraham.ac.uk/projects/fastqc/ |
| Software, algorithm | TopHat | (*Trapnell et al., 2009*) | | http://ccb.jhu.edu/software/tophat/index.shtml |
| Software, algorithm | Samtools | (*Li et al., 2009*) | | http://www.htslib.org/ |
| Software, algorithm | HOMER | (*Heinz et al., 2010*) | | http://homer.ucsd.edu/homer/ |
| Software, algorithm | DESeq2 | (*Love et al., 2014*) | | https://bioconductor.org/packages/release/bioc/html/DESeq2.html |
| Software, algorithm | edgeR | (*Robinson et al., 2010*) | | https://bioconductor.org/packages/release/bioc/html/edgeR.html |
| Software, algorithm | featureCounts | (*Liao et al., 2014*) | | http://bioinf.wehi.edu.au/featureCounts/ |
| Software, algorithm | WGCNA | (*Langfelder and Horvath, 2008*) | | https://cran.r-project.org/web/packages/WGCNA/index.html |
| Software, algorithm | Trim Galore | other | | https://www.bioinformatics.babraham.ac.uk/projects/trim_galore/ |
| Software, algorithm | Bismark | (*Krueger and Andrews, 2011*) | | https://www.bioinformatics.babraham.ac.uk/projects/bismark/ |
| Software, algorithm | methylKit | (*Akalin et al., 2012*) | | https://bioconductor.org/packages/release/bioc/html/methylKit.html |
| Software, algorithm | STRING | (*Szklarczyk et al., 2019*) | | https://string-db.org/ |
| Software, algorithm | MeV | other | | https://sourceforge.net/projects/mev-tm4/ |
| Software, algorithm | ggplot2 | (*Wickham, 2016*) | | https://github.com/tidyverse/ggplot2 |
| Software, algorithm | qqman | (*Turner, 2014*) | | https://github.com/stephenturner/qqman |
| Software, algorithm | dplyr | (*Wickham et al., 2018*) | | https://dplyr.tidyverse.org |
| Software, algorithm | Ingenuity Pathway Analysis | Qiagen | | https://www.qiagenbioinformatics.com/products/ingenuity-pathway-analysis |
| Software, algorithm | ImageJ2 (Fiji) with trackmate | NIH | | https://imagej.net/ImageJ2 |
| Software, algorithm | BioVenn | (*Hulsen et al., 2008*) | | http://www.biovenn.nl/ |
| Software, algorithm | Prism | GraphPad Software | | https://www.graphpad.com/scientific-software/prism/ |

## Generation of clonal cell lines and cell culture

Immortalized mouse ear fibroblast cells from male mice carrying PER2::LUC*sv* bioluminescence reporter were maintained in DMEM (Corning) supplemented with 10% fetal bovine serum (FBS). To generate clonal cell lines, cells were diluted and seeded at a density of ~30 cells per 96-well plate with conditioned medium. Each well was monitored on a daily basis to make sure only single colonies were picked. 20 clonal cell lines were randomly selected and cultured continuously for 20 generations (3 days/generation) to verify stability of circadian period. Primary mouse embryonic fibroblast (MEF) cells carrying PER2::LUC*sv* bioluminescence reporter were isolated from 13.5 day mouse

embryos. NIH3T3 cells stably expressing *Per2* E-box (E2)-driven luciferase bioluminescence reporter were established by lentivirus transduction followed by blasticidin selection. Our cell line stocks have all tested negative for mycoplasma contamination. For authentication of cell lines, as described below, two clonal cell lines, #33 and #114, were sequenced by whole exome sequencing; and a total of 34 clones and subclones were assessed by RNA-seq and were found to be valid.

## Bioluminescence imaging and data analysis

To measure luminescence rhythms from 35 mm culture dishes, confluent cells were synchronized with 100 nM dexamethasone for 2 hr, then changed to HEPES-buffered recording medium containing 2% FBS (*Welsh et al., 2004*), and loaded into a LumiCycle luminometer (Actimetrics). The period was analyzed with LumiCycle Analysis program (Actimetrics). All LumiCycle period analysis results shown in this paper were averages of ≥3 experiments. Baseline-subtracted signals were exported to Excel to generate bioluminescence traces.

For single-cell imaging, cells were changed to recording medium containing 2% B27% and 1% FBS without dexamethasone synchronization. An inverted microscope (Leica DM IRB) in a heated lucite chamber custom-engineered to fit around the microscope stage (Solent Scientific, UK) kept the cells at a constant 36°C was mounted on an anti-vibration table (TMC) equipped with a 10X objective. A cooled CCD camera with backside illuminated E2V CCD 42–40, 2048 × 2048 pixel, F-mount adapter, −100°C cooling (Series 600, Spectral Instruments) was used to capture the luminescence signal at 30 min intervals, with 29.6 min exposure duration, for at least 12 days. 8 × 8 binning was used to increase the signal-to-noise ratio. The bioluminescence signal of each single cell, outlined with a region of interest (ROI), was tracked using ImageJ (*Schindelin et al., 2012*; *Rueden et al., 2017*) with the Trackmate plugin (*Tinevez et al., 2017*) and analyzed as described previously (*Li et al., 2020*).

## Next generation sequencing and data analysis

For exome sequencing, two clonal cell lines #33 and #114 were sequenced representing short period and long period clones, respectively. Genomic DNA was purified using a ChargeSwitch gDNA Mini Tissue Kit (Invitrogen). Libraries were made using the SureSelect$^{XT}$ Reagent Kit (Agilent) following the manufacturer's instruction. All reads were mapped to mm10 genome assembly. We used Haplo-typeCaller and UnifiedGenotyper from GATK to call variants and the results were the union of both callers. SnpEff was used to annotate variants. Results were further filtered as follows: threshold GQ $\geq$ 20, total counts $\geq$ 8, and alternate frequency (defined as the ratio of alternates to total counts)≥30%.

For RNA-seq, cells were collected at two time-points after synchronization: the first peak (T1) and the following trough (T2) based on LumiCycle recording. At each time-point, we collected 2 replicates for 10 clonal cell lines and 1 replicate for 24 subclones. RNA was isolated using TRIzol (Life technologies), and libraries were prepared as described previously (*Takahashi et al., 2015*). Raw reads were tested for quality using FastQC. The resulting reads were mapped to mm10 annotation from UCSC using TopHat (*Trapnell et al., 2009*). The output BAM file was then filtered for uniquely mapped reads using Samtools (*Li et al., 2009*), and RPKM calculations were performed using analyzeRepeats.pl of HOMER suite (*Heinz et al., 2010*).

The average RPKM value for each gene was calculated separately for each of the six groups (SP, LP, SSP, LSP, SLP, LLP). To identify significant DE genes, the list was further filtered based on expression level. Only genes for which the maximum average RPKM value among six groups was greater than 0.5 were preserved. Differential gene expression analysis was carried out with DESeq2 (*Love et al., 2014*) and edgeR (*Robinson et al., 2010*) using a raw read counts matrix generated with featureCounts tool (*Liao et al., 2014*). Genes with FDR < 0.05 were deemed significant. Results from both programs were combined to generate a final DE gene list. Pearson correlation coefficient between circadian period length and gene expression was calculated across all 88 samples (including replicates and different time-points) in Excel. P-value was adjusted using Benjamini-Hochberg (BH) method, and FDR < 0.05 was considered as significant. The overlaps between significant DE genes and period-correlated genes were defined as period-correlated DE genes. Multidimensional scaling (MDS) analysis with Euclidean distance was performed using edgeR. Ingenuity Pathway Analysis

(Qiagen) was used to identify the pathways associated with period-correlated DE genes, using all expressed 22,786 genes (average RPKM >0) as a reference set.

For DNA methylation sequencing, cells were collected at the first peak (T1) after synchronization. Each clone included two replicates. DNA was purified using a PureLink Genomic DNA Mini Kit (Invitrogen). Libraries were made using the Premium Reduced Representation Bisulfite Sequencing (RRBS) Kit (Diagenode) following the manufacturer's instruction. Raw reads were tested for quality using FastQC and trimmed with Trim Galore. The trimmed reads were aligned to mm10 using Bismark (*Krueger and Andrews, 2011*). The CpG reports from Bismark methylation extractor were then analyzed using methylKit (*Akalin et al., 2012*). We used default settings to discard bases that had coverage below 10X and/or more than 99.9th percentile of coverage in each sample. Differentially methylated regions (DMR) were identified using a tiling window of 1,000 bp and a step size of 1,000 bp comparing SP and LP group. Clone#44 was excluded for DMR analysis because of the outlying clustering (*Figure 5—figure supplement 1*). Overdispersion correction with Fisher's extract test was applied. P-value was adjusted with BH method. DMRs with FDR < 0.05 and methylation difference >25% were considered as significant. Genes with significant DMRs located either in the gene body or 5 kb upstream of the transcription start site (TSS) were considered as DMR-associated genes. Principal component analysis (PCA) was performed using methylKit. All sequencing was performed by the UTSW McDermott Sequencing Core Facility.

## Weighted Gene Co-expression Network Analysis (WGCNA)

Weighted gene co-expression network analysis was performed using WGCNA package (*Langfelder and Horvath, 2008*). Only genes for which the maximum average RPKM value among six groups was greater than 0.5 RPKM were used. A soft-threshold power was automatically calculated to achieve approximate scale-free topology ($R^2$ >0.85). Networks were constructed with `blockwiseModules` function with biweight midcorrelation (bicor). We used `corType = bicor`, `networkType = signed`, `TOMtype = signed`, `TOMDenom = mean`, `maxBlockSize = 16000`, `mergingThresh = 0.10`, `minCoreKME = 0.4`, `minKMEtoStay = 0.5`, `reassignThreshold = 1e-10`, `deepSplit = 4`, `detectCutHeight = 0.999`, `minModuleSize = 100`, `power = 26`. The modules were then determined using the dynamic tree-cutting algorithm. Deep split of 4 was used to split more aggressively the data and create more specific modules. Spearman's rank correlation was used to compute module eigengene – covariates associations. Gene set enrichment applied for module – period-correlated DE genes was performed using a Fisher's exact test in R with the following parameters: `alternative = 'greater'`, `conf.level = 0.99`. The PPI network was generated using STRING without textmining, and the minimum required interaction score was 0.7 (*Szklarczyk et al., 2019*).

Gene Knockdown Assay shRNA sequences were cloned into pLKO.1-TRC vector (gift from David Root, Addgene plasmid # 10878) (*Moffat et al., 2006*). Scramble shRNA (5' -CCTAAGGTTAAG TCGCCCTCG- 3') was used as control. Lentiviruses were produced using HEK293T cells as described previously (*Huang et al., 2012*). Viruses were harvested twice after transfection, at 48 and 72 hr, to infect fibroblasts. Forty-eight hours after first infection, cells were synchronized and loaded for Lumi-Cycle analysis. RNA was extracted at the first peak after synchronization to check knockdown efficiency via qPCR. Average of three reference genes (*Gapdh*, *Hprt* and *Ywhaz*) served as internal control. See *Supplementary files 2* and *3* for shRNA target sequences and primer sequences, respectively.

## Drug treatment

The EIF2 signaling pathway activator halofuginone (Sigma-Aldrich) was dissolved in DMSO as 10 mM stock and used at 50 nM. Tunicamycin (Sigma-Aldrich) was dissolved in DMSO as 5 mg/ml stock and used at 5 µg/ml. Cells were treated for 4 hr and 6 hr, respectively, before loading for LumiCycle analysis. DNMT inhibitor SGI-1027 (Sigma-Aldrich) was dissolved in DMSO as 200 mM stock and used at 10 µM. Zebularine (Sigma-Aldrich) was dissolved in water as 200 mM stock and used at 50 µM or 100 µM. The parent culture was continuously treated for up to 60 days and split when necessary. MEFs and NIH3T3 cells were treated for 3 days.

## Quantification and statistical analysis

Statistical analysis of single-cell imaging was performed with a Python code as described previously (*Li et al., 2020*). Student's T-test and two tailed F-test were performed in Excel. P-values were adjusted using Benjamini-Hochberg (BH) method. Two-way ANOVA analysis with multiple comparisons via Tukey test was performed using GraphPad Prism. Heatmaps for single-cell imaging analysis and gene expression were generated using MeV based on z-score. GraphPad prism was used to generate heatmaps for T-test and F-test based on log transformed q-value. Volcano plot was generated in R using ggplot2 (*Wickham, 2016*). Venn diagrams were generated using BioVenn (*Hulsen et al., 2008*). Manhattan plots were generated in R using qqman (*Turner, 2014*). Quadrant plots were generated using dplyr package in R.

## Acknowledgements

This research was supported by the Howard Hughes Medical Institute. All bioinformatics analyses were carried out on Stampede2 cluster of TACC at UT Austin. The authors would like to thank all Takahashi lab members, Dr. Carla B Green, and Dr. Shin Yamazaki for helpful discussions, and the McDermott Bioinformatics Lab at UT Southwestern Medical Center for their bioinformatics support. JST is an Investigator in the Howard Hughes Medical Institute.

## Additional information

### Funding

| Funder | Author |
| --- | --- |
| Howard Hughes Medical Institute | Joseph S Takahashi |

The funders had no role in study design, data collection and interpretation, or the decision to submit the work for publication.

### Author contributions

Yan Li, Conceptualization, Formal analysis, Investigation, Visualization, Methodology, Writing - original draft, Writing - review and editing; Yongli Shan, Formal analysis, Investigation, Methodology; Gokhul Krishna Kilaru, Data curation, Formal analysis; Stefano Berto, Guang-Zhong Wang, Genevieve Konopka, Formal analysis; Kimberly H Cox, Visualization, Writing - review and editing; Seung-Hee Yoo, Shuzhang Yang, Resources, Methodology; Joseph S Takahashi, Conceptualization, Funding acquisition, Writing - review and editing

### Author ORCIDs

Yan Li (iD) https://orcid.org/0000-0003-4962-9910
Guang-Zhong Wang (iD) https://orcid.org/0000-0001-6432-8310
Kimberly H Cox (iD) https://orcid.org/0000-0002-7097-9563
Genevieve Konopka (iD) http://orcid.org/0000-0002-3363-7302
Joseph S Takahashi (iD) https://orcid.org/0000-0003-0384-8878

### Decision letter and Author response

Decision letter https://doi.org/10.7554/eLife.54186.sa1
Author response https://doi.org/10.7554/eLife.54186.sa2

## Additional files

### Supplementary files

- Supplementary file 1. Exome Sequencing of Clone#33 and Clone#114.
- Supplementary file 2. shRNA Target Sequences.
- Supplementary file 3. Primer Sequences.

• Transparent reporting form

### Data availability

RNA Sequencing data have been deposited in GEO under accession codes: GSE132663 and GSE132665. Exome sequencing data have been deposited in SRA under accession number: PRJNA548837. All data generated or analyzed during this study are included in the manuscript and supporting files. Source data have been provided for Figures 2 and 4.

The following datasets were generated:

| Author(s) | Year | Dataset title | Dataset URL | Database and Identifier |
|---|---|---|---|---|
| Li Y, Takahashi JS | 2019 | Transcriptional Profiling of Clonal Cell Lines with Different Circadian Period | https://www.ncbi.nlm.nih.gov/geo/query/acc.cgi?acc=GSE132663 | NCBI Gene Expression Omnibus, GSE132663 |
| Li Y, Takahashi JS | 2019 | RRBS Profiling of Clonal Cell Lines with Different Circadian Period | https://www.ncbi.nlm.nih.gov/geo/query/acc.cgi?acc=GSE132665 | NCBI Gene Expression Omnibus, GSE132665 |
| Li Y, Takahashi JS | 2019 | Exome-seq of mouse immortalized ear fibroblast clonal cell lines with different circadian periods | http://www.ncbi.nlm.nih.gov/bioproject/?term=PRJNA548837 | NCBI BioProject, PRJNA548837 |

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
