## [Decision Letter]

**Acceptance summary:**

The manuscript from the Takahashi laboratory uses a culture model to investigate mechanisms underlying cell-to-cell variations in circadian period. The authors find that periodicity is a stable, heritable phenotype, and differences in period correlate with changes in gene expression. DNA methylation is implicated as a significant source of period variability across cells/clones.

**Decision letter after peer review:**

Thank you for submitting your article "Epigenetic Inheritance of Circadian Period in Clonal Cells" for consideration by *eLife*. Your article has been reviewed by three peer reviewers, and the evaluation has been overseen by a Reviewing Editor and Catherine Dulac as the Senior Editor. The following individuals involved in review of your submission have agreed to reveal their identity: Salvador Aznar Benitah (Reviewer #2); Michael Rosbash (Reviewer #3).

The reviewers have discussed the reviews with one another and the Reviewing Editor has drafted this decision to help you prepare a revised submission.

Summary:

The manuscript from the Takahashi laboratory investigates the mechanism underlying differences in circadian period across cultured fibroblasts. The authors generated clonal populations of varying circadian period and find that periodicity is a stable, heritable phenotype. Transcriptomic analysis of short and long period clones identified genes whose expression correlates with period, and knockdown of most of these genes affects circadian period, albeit not necessarily in the direction predicted by the correlational analysis. Finally, differences in methylation are proposed as the basis of period alterations.

Essential revisions:

The reviewers found the study of interest, but noted a few concerns. First, it is critical that the authors establish a/the mechanism that accounts for the differences in period across clones. As currently presented, genes showing a correlation with period are knocked down in the parent culture and, as noted above, their effects on period are in the opposite direction from what one would predict. It is also surprising that so many of them affect period, raising the question of whether uncorrelated genes would also show effects on period. Reviewers recommend that knockdown (or perhaps over-expression) be conducted in isolated clones or sub-clones to "rescue" the differences in period. Inclusion of uncorrelated genes would demonstrate specificity. If the opposite effects on period result from multiple genes acting together, as claimed by the authors, then they should be able to manipulate pathways to nail the mechanism. Finally, to better integrate the methylation analysis, it would be good to see changes in methylation linked to the specific genes manipulated, and not to correlated networks in general.

Knockdown of DNMTs in clones would also be informative.

Related to the above point, correlation and knockdown data should be included in the main text (and not the supplement).

The second major point raised by reviewers concerns the relevance of these findings to other systems. Ideally, this would be demonstrated in vivo, or at least in primary cells, but recognizing the amount of time these experiments would take, examples in other cell types would suffice.

---

## [Author Response]

Essential revisions:The reviewers found the study of interest, but noted a few concerns. First, it is critical that the authors establish a/the mechanism that accounts for the differences in period across clones. As currently presented, genes showing a correlation with period are knocked down in the parent culture and, as noted above, their effects on period are in the opposite direction from what one would predict. It is also surprising that so many of them affect period, raising the question of whether uncorrelated genes would also show effects on period. Reviewers recommend that knockdown (or perhaps over-expression) be conducted in isolated clones or sub-clones to "rescue" the differences in period. Inclusion of uncorrelated genes would demonstrate specificity.

Thank you for this comment. It is true that knockdown (KD) of some genes yielded effects on period that were opposite of what we would have predicted, while KD of others resulted in the expected changes in period. We have also now included data from two additional period-correlated genes whose KD did not change period length (Figure 3A-F). Altogether, our results suggest that the period-correlated DE genes identified here are a promising candidate pool; however, while each candidate gene may need to be validated in the future, it is not possible or reasonable to validate “all” of the candidate genes in this paper. We have rigorously validated 14 gene knockdowns and 12 of these either lengthened or shortened the period length, demonstrating that the majority (86%) of candidate genes change period length.

While we attempted to complete additional experiments prior to the laboratory shutdown during the COVID-19 pandemic, we were not successful in achieving knockdown of additional uncorrelated genes and were unable to finish any additional analysis of uncorrelated gene KD’s.

For period-correlated gens, we agree with the suggestion to show KD results from the isolated clones and, as recommended, we have now added KD data from 10 clones for 5 representative genes (Figure 3G).

If the opposite effects on period result from multiple genes acting together, as claimed by the authors, then they should be able to manipulate pathways to nail the mechanism.

This is a fair point. We have now added data from cells treated with two drugs that can activate the EIF2 signaling pathway (Figure 4D). A previous publication (Pathak et al., 2019) has a full series of data using both KO mice and drugs to prove that the EIF2 signaling pathway can regulate circadian period. Instead of repeating these experiments, we utilized two additional activators (not used in that paper). Thus, we are able to show that our data are consistent with the literature (Pathak et al., 2019) in showing that the EIF2 signaling pathway is one pathway that can influence period. In addition, we have already shown that KDs of the DNA methylation pathway regulate period, consistent with the reviewer’s point above concerning “manipulating pathways”.

Finally, to better integrate the methylation analysis, it would be good to see changes in methylation linked to the specific genes manipulated, and not to correlated networks in general.Knockdown of DNMTs in clones would also be informative.Related to the above point, correlation and knockdown data should be included in the main text (and not the supplement).

Thank you for this suggestion. In our study, most DMR associated DE genes are in “on” or “off” status, while the previously identified period-correlated DE genes are expressed in a gradient. That is why we did not see significant methylation changes associated with those genes manipulated in Figure 3. We have now added additional gene KD data for four DMR-associated genes. Of these, three out of four have significant results consistent with predictions (Figure 6D).

In addition, we have added experiments with *Dnmt1* and *Dnmt3a* KD in 10 clones (Figure 7C).

Again, we would argue that “correlated networks” address the very point that the reviewer made in the previous section that we “should be able to manipulate pathways”.

The second major point raised by reviewers concerns the relevance of these findings to other systems. Ideally, this would be demonstrated in vivo, or at least in primary cells, but recognizing the amount of time these experiments would take, examples in other cell types would suffice.

We agree it would be ideal to show knockdown in other cell lines. We had planned these experiments, using primary MEF cells with PER2::LUC*sv* and a NIH3T3 cell line carrying an E2-box-luc, but were unable to complete them before the COVID-19 shutdown. However, we were able to complete experiments using two DNA methylation inhibitors in these two additional cell lines. Both of these additional cell types yielded results similar to our original cell lines (Figure 7E). Thus, we can now show that we have extended these findings to other systems (cell types).